# Can We Use Satellite-Based Soil-Moisture Products at High Resolution to Investigate Land-Use Differences and Land–Atmosphere Interactions? A Case Study in the Savanna

**Carlos Román-Cascón [1,2,*]**, **Marie Lothon [2]**, **Fabienne Lohou [2]**, **Nitu Ojha [3]**, **Olivier Merlin [3]**, **David Aragonés [4]**, **María P. González-Dugo [5]**, **Ana Andreu [5]**, **Thierry Pellarin [6]**, **Aurore Brut [3]**, **Ramón C. Soriguer [7]**, **Ricardo Díaz-Delgado [4]**, **Oscar Hartogensis [8]** and **Carlos Yagüe [9]**

[1] Centre National d'Études Spatiales (CNES), 31400 Toulouse, France
[2] Laboratoire d'Aerologie, CNRS, Université de Toulouse, 31400 Toulouse, France;
marie.lothon@aero.obs-mip.fr (M.L.); fabienne.lohou@aero.obs-mip.fr (F.L.)
[3] CESBIO, Université de Toulouse, CNES/CNRS/INRAE/IRD/UPS, 31400 Toulouse, France;
ojhan@cesbio.cnes.fr (N.O.); olivier.merlin@cesbio.cnes.fr (O.M.); aurore.brut@cesbio.cnes.fr (A.B.)
[4] Remote Sensing & GIS Lab. Estación Biológica de Doñana-CSIC, 41092 Sevilla, Spain;
daragones@ebd.csic.es (D.A.); rdiaz@ebd.csic.es (R.D.-D.)
[5] IFAPA. Avd. Menéndez Pidal s/n, 14071 Córdoba, Spain;
mariap.gonzalez.d@juntadeandalucia.es (M.P.G.-D.); anandreum@posteo.net (A.A.)
[6] CNRS, IRD, Univ. Grenoble Alpes, Grenoble INP, IGE, F-38000 Grenoble, France;
thierry.pellarin@univ-grenoble-alpes.fr
[7] Estación Biológica de Doñana, CSIC, 41092 Sevilla, Spain; soriguer@ebd.csic.es
[8] Meteorology and Air Quality Section, Wageningen University, 6700AA Wageningen, The Netherlands;
oscar.hartogensis@wur.nl
[9] Departamento de Física de la Tierra y Astrofísica. Universidad Complutense de Madrid, 28040 Madrid, Spain; carlos@ucm.es
[*] Correspondence: carlosromancascon@ucm.es

**Abstract:** The use of soil moisture (SM) measurements from satellites has grown in recent years, fostering the development of new products at high resolution. This opens the possibility of using them for certain applications that were normally carried out using in situ data. We investigated this hypothesis through two main analyses using two high-resolution satellite-based soil moisture (SBSM) products that combined microwave with thermal and optical data: (1) The Disaggregation based on Physical And Theoretical scale Change (DISPATCH) and, (2) The Soil Moisture Ocean Salinity-Barcelona Expert Center (SMOS-BEC Level 4). We used these products to analyse the SM differences among pixels with contrasting vegetation. This was done through the comparison of the SM measurements from satellites and the measurements simulated with a simple antecedent precipitation index (API) model, which did not account for the surface characteristics. Subsequently, the deviation of the SM from satellite with respect to the API model (bias) was analysed and compared for contrasting land use categories. We hypothesised that the differences in the biases of the varied categories could provide information regarding the water retention capacity associated with each type of vegetation. From the satellite measurements, we determined how the SM depended on the tree cover, i.e., the denser the tree cover, the higher the SM. However, in winter periods with light rain events, the tree canopy could dampen the moistening of the soil through interception and conducted higher SM in the open areas. This evolution of the SM differences that depended on the characteristics of each season was observed both from satellite and from in situ measurements taken beneath a tree and in grass on the savanna landscape. The agreement between both types of measurements

highlighted the potential of the SBSM products to investigate the SM of each type of vegetation. We found that the results were clearer for DISPATCH, whose data was not smoothed spatially as it was in SMOS-BEC. We also tested whether the relationships between SM and evapotranspiration could be investigated using satellite data. The answer to this question was also positive but required removing the unrealistic high-frequency SM oscillations from the satellite data using a low pass filter. This improved the performance scores of the products and the agreement with the results from the in situ data. These results demonstrated the possibility of using SM data from satellites to substitute ground measurements for the study of land–atmosphere interactions, which encourages efforts to improve the quality and resolution of these measurements.

**Keywords:** soil moisture; satellite data; land use; heterogeneity; savanna; DISPATCH; SMOS-BEC

## 1. Introduction

Soil moisture (SM) is defined as an essential climate variable by the European Space Agency (ESA) [1]. This variable is key for the water [2], energy [3], and carbon [4,5] cycles. Therefore, precise knowledge of the amount of water in the soil is crucial for hydrological, e.g., [6], meteorological, e.g., [7], climatological, e.g., [8] and ecological modelling, including pathological and epidemiological applications, e.g., [9,10]. However, this variable exhibits important horizontal heterogeneity at different scales [11], even within a few meters [12,13]. In situ measurements normally provide accurate soil-moisture dynamics over a single point. However, in heterogeneous areas, the SM can evolve in a different way in nearby measurements with contrasting soil and/or surface characteristics, such as the vegetation cover, e.g., [14–16]. The spatial heterogeneity in the vegetation cover may lead to different SM dynamics among patches [17]. This is related to the different capacity of each type of vegetation for transpiration, water retention in the soil (avoiding evaporation), interception, water use from shallower or deeper layers [18], different seasonal plant activity, or soil litter production. The relative importance of each of these physical processes is sometimes difficult to determine for each vegetation type and site, which generates uncertainty about the interpretation of the observed SM dynamics [19].

These uncertainties can impact other practical applications. One example is the evaluation of land-surface models in heterogeneous areas. In some cases, accurate and extensive in situ measurements are available over different land cover types, as in field campaigns, e.g., [20]. This enables us to analyse how moisture transfers from the land to the atmosphere behave over different surfaces, as well as to fairly compare models and observations. However, in situ measurements over different surfaces are not always available and, even in the case of field campaigns, it is difficult to determine area-averaged representative values to be compared with model outputs at their typical resolutions.

In this context, analyzing SM from space seems a good alternative to complement the measurements performed on the ground [21]. This is more evident now after the launching of satellites specifically designed for this aim, like SMOS (Soil Moisture and Ocean Salinity, [22]) or SMAP (Soil Moisture Active Passive, [23]), both based on the radiometric retrievals in the microwave (MW) L-band at resolutions of around 40 km. However, while this resolution is useful for many global and regional applications of different natures, e.g., [24–28], it is not enough to study land-vegetation-atmosphere interactions at the resolution of typical land-surface and numerical weather prediction models, i.e., a few kilometres.

Given the limitations of the ground-based and the satellite SM measurements for resolutions of around 1–2 km, the development of new satellite-based soil-moisture (SBSM) products at higher resolutions (1 km and currently increasing) seems promising. In this paper two SBSM products based on SMOS data were analysed: (1) The Disaggregation based on Physical And Theoretical

scale Change (DISPATCH, [29,30]) and, (2) The SMOS-Barcelona Expert Center (SMOS-BEC, [31,32]). These two products are based on the downscaling of SM values provided by the SMOS satellite, which was launched in 2009 by the ESA. The downscaling is possible thanks to the triangular relationship between these SM data together with the normalised difference vegetation index (NDVI) and land surface temperature (LST) information. Although the physical basis of both products is the evapotranspiration process in moisture-limited conditions, each product is based on algorithms with different particularities (detailed in Section 2.2). These high-resolution (1 km) products are quite attractive for researchers working in disciplines where the water cycle is crucial, such as hydrologists, meteorologists, climatologists, ecologists, or agronomists [33]. Among the potential advantages of these products are the global coverage or the adequate spatial resolution for specific purposes, which could imply a better representativity of areas of interest (crop fields, forests, prairies, irrigation zones, etc.). However, some hesitations exist about using them for land–atmosphere research, such as the low temporal resolution, the possible uncertainties in their absolute values or the typical noise observed in the satellite-based SM signals [34]. In this sense, the objective of this paper was to analyse the information provided by the described SBSM products over different land covers and at the pixel scale for this type of research.

In this paper we tried to answer two scientific questions: (1) Can we identify SM differences (and their evolution through the year) due to contrasting land cover (vegetation) using high-resolution SBSM products? and; (2) Can these products be used to analyse land-atmosphere interactions at the local-regional scale? Although many works have analysed spatial differences in SM content due to the effects of the different types of vegetation, e.g., [18,19], the analysis from satellite data has not been exploited yet [35]. This could open an alternative to the common use of in situ data for these aims.

The study was performed in an oak-grass savanna landscape in the south of Spain. This landscape is appropriate for this type of analysis because two markedly different land covers are present: grass and savanna trees. Therefore, the results can be distinguished for each vegetation type, while the absence of additional land cover categories (or very small proportions) eases the analysis. While the changes in SM spatial variability due to the vegetation type are less evident in sites without SM limitation, these effects are more apparent in sites where this variable is limiting [19], as is the case of the semi-arid region analysed here. Furthermore, this type of agrosilvopastoral landscape, covering more than 3 million ha in Europe, is key for rural economies but very sensitive to strong drought periods [36], which encourages such a particular study. Finally, although it was not the main objective of this work, the intrinsic SBSM product evaluation included in this work could be useful to contribute to the scientific development and improvement of the SBSM products analysed.

The paper has been organized as follows: in Section 2, we present the methods and data used; in Section 3, we present the results obtained, divided into the two scientific questions raised above; and in Section 4, we provide a brief discussion and a summary of the main findings.

## 2. Methods and Data

The first part of this work focuses on the relationship between land use (LU) and SM from satellite data in 2015 in the area of study, located in southern Spain. The first step was to simulate the SM with a simple antecedent precipitation index (API) model. Then, we calculated the biases of the SBSM products when compared to the simulated SM from the API. These biases were then analysed considering the type of LU of each pixel, focusing on their relative differences. We hypothesised that these biases contained information about the real water content in the soil for each vegetation cover. To avoid the differences due to the different types of soil, only those pixels with the same type of soil were compared (loam, which was the dominant soil in the area of study).

The second part of this work shows a comparison of the previous information from satellites with the in situ data obtained in an experimental site situated over the savanna area, using the SM data measured in the grass and beneath a tree. Finally, we tried to determine whether the interactions between the surface and the atmosphere can also be investigated using data from satellites instead of

the typical in situ data used. The different datasets employed in this study are summarised in Table 1 and explained in the next subsections.

**Table 1.** Brief summary of the different types of data used in this work. The original resolution of the products and their more important re-processing is included in the info column.

| Type of data | Product | Info |
|---|---|---|
| Satellite soil moisture | DISPATCH (1 km) <br> SMOS-BEC L4 (1 km) <br> SMOS (25 km) | SMOS SM; MODIS LST and NDVI [29,30]. <br> SMOS SM; ECMWF LST and MODIS NDVI [31,32]. <br> L3 product produced by CATDS [22]. |
| Land use | SYPNA (10 m) | Re-gridded to 1 km. |
| Rainfall (for API) <br> 2-m temperature (for API) | SPAIN-02 v5 (0.1°) <br> SPAIN-02 v5 (0.1°) | Gridded database [37–39]. <br> Gridded database [37–39]. |
| Soil type | USDA (0.08°) | USDA 16-cat database, re-gridded to 3 km [40]. |
| In situ soil moisture <br> In situ latent heat flux <br> In situ net radiation | EnviroSCAN probes (10–30–50 cm) <br> LICOR (18 m agl) <br> NR01 radiometer (18 m agl) | Grass and tree sites [41,42]. <br> Measurements above tree [41,42]. <br> Measurements above tree [41,42]. |

### 2.1. Region Analysed and In Situ Data

The area of study covers a region of approximately 3500 km$^2$ in southern Spain (yellow rectangle in Figure 1a, zoomed in Figure 1b). This region is dominated by a Mediterranean climate with a long dry season, and is, to a large extent, covered by savanna (*dehesa*), with oak trees (mainly *Quercus Ilex*) and grass in variable proportions. The SM retrievals from satellite are well-known to work better in sites with low vegetation densities and, in the case of DISPATCH and SMOS-BEC, in regions without too much cloudiness since their algorithms use thermal and optical data only available under clear-sky conditions. Both conditions (no dense vegetation and clear sky weather) are often fulfilled in the region.

A total of 3500 pixels of $1 \times 1$ km were analysed in this work from satellite data; however in situ measurements taken at the local scale were also used to compare with the satellite measurements and to perform some of the analyses. These measurements were taken over the *Santa Clotilde* experimental site (Figure 2), the location of which is shown with a yellow circle in Figure 1b (38°12′N; 4°17′W, 736 m above sea level). In this site, the proportions of trees and grass is estimated to be 20% and 80% respectively (the site is identified as dense savanna from the LU database). This site is managed by the Andalusian Institute for Agricultural and fisheries Research and Training (IFAPA) to monitor ecological data in the savanna region for research and management purposes (note how this type of landscapes covers a large part of the surface of southern European countries).

The site contains several instruments, including a total of six SM probes (EnviroSCAN, Sentek Technologies, Stepney, Australia) at two grazing exclusion enclosures (over open grassland and under a holm-oak tree) and at three different depths (10, 30, and 50 cm). An open-path $CO_2$/$H_2O$ Gas Analyzer LICOR-7500 and a 3D sonic anemometer (model CSAT3 Campbell Scientific Inc.) are also installed in an Eddy-Covariance (EC) tower at 18 meters above ground level (agl), providing evapotranspiration and carbon measurements and operating from 2012 to the present (2020). The set up included the instruments required for continuously measuring all surface energy balance components in addition to the turbulent fluxes (sensible and latent heat flux), these being the radiation budget (four-component net radiometer model NR01, Hukseflux Thermal Sensors) and the heat flux transport across the surface soil (model HFP01, Huseflux Thermal Sensors). More information regarding the study area and the equipment and data processing can be found in [41–43].

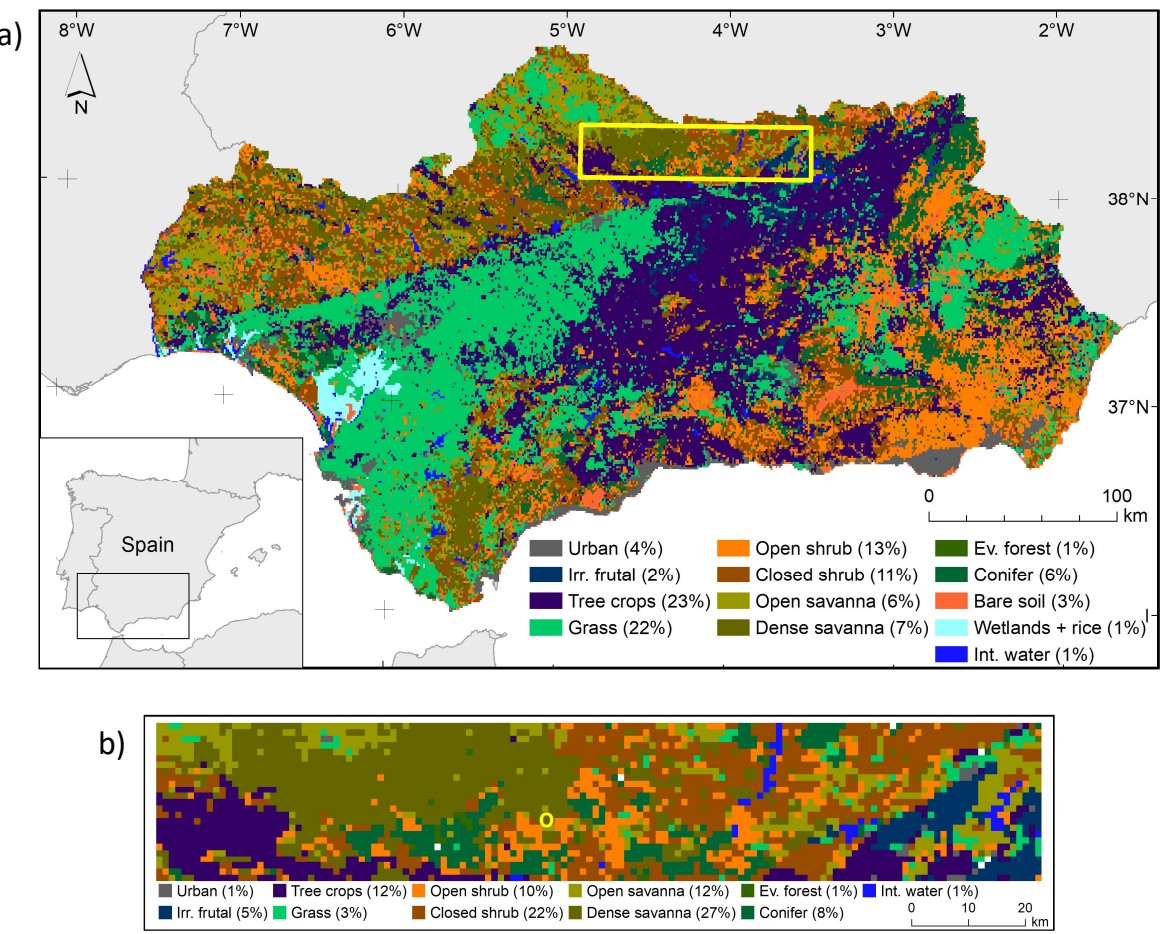

**Figure 1.** (**a**) Dominant land use (LU) in Andalusia (south of Spain), the yellow rectangle shows the area analysed in the Sierra Morena region. (**b**) Zoom of the area analysed in this study, marked with a yellow rectangle in (**a**). The *Santa Clotilde* experimental site is indicated with a yellow circle.

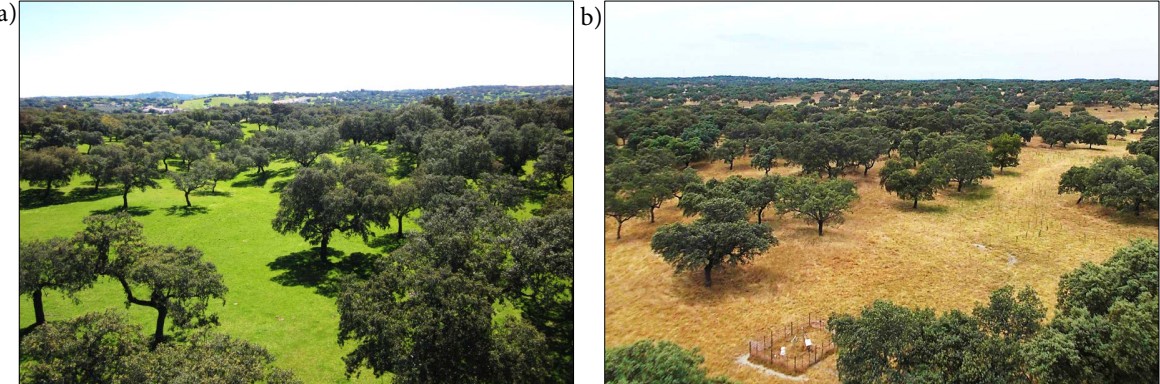

**Figure 2.** Pictures taken from the tower installed at the *Santa Clotilde* site in spring (7 April 2014) (**a**) and in summer (21 June 2013) (**b**).

*2.2. Satellite-Based Soil-Moisture Products*

1. **DISPATCH (1 km)**: The Disaggregation based on Physical And Theoretical scale Change (DISPATCH) [29,30] approach relates the SMOS SM to the soil evaporative efficiency (SEE,

ratio of actual to potential soil evaporation) derived from the Moderate Resolution Imaging Spectroradiometer (MODIS) LST and NDVI data. A digital elevation model was also used to correct the LST for elevation effects prior to the SEE estimation. A SEE model is first calibrated at the SMOS pixel scale using the SMOS SM and aggregated MODIS-derived SEE. The DISPATCH disaggregated SM within the selected SMOS pixel is then expressed as a Taylor series expansion around the SMOS SM given the calibrated SEE (SM) model and the 1 km SEE difference to the SMOS resolution.

2. **SMOS-BEC L4 (1 km)**: The SMOS-Barcelona Expert Center (SMOS-BEC) [31,32] disaggregation approach relates the SMOS SM to ancillary data composed of the normalized LST from the European Centre for Medium-Range Weather Forecasts (ECMWF), the MODIS NDVI and the normalized SMOS brightness temperatures at three distinct incidence angles in the vertical and horizontal polarizations. The five empirical coefficients of this relationship were first estimated by aggregating the ECMWF and MODIS data at SMOS resolution (i.e., 25 km) and by applying the equation to several SMOS pixels (a window composed of nine pixels in [32]) around the selected SMOS pixel. The SMOS-BEC disaggregated SM within the moving window was then estimated by applying the calibrated equation to the 1 km resolution data. The SMOS brightness temperatures used as inputs were previously re-sampled at a 1 km resolution to fit the output downscaling resolution. As DISPATCH and SMOS-BEC have different grid coordinates, the 1-km grid of SMOS-BEC was interpolated to the same 1-km grid as DISPATCH using the nearest neighbour interpolation method, in order to compare both with the LU data prepared in the same reference grid.

3. **SMOS**: In order to compare the performance of the downscaling products when compared to in situ data, the 25-km resolution (Equal-Area Scalable Earth Grid (EASE)) level-3 RE04 SMOS dataset provided by CATDS (*Centre Aval de Traitement des Données SMOS*) was used, obtained at the pixel of the experimental site. Three different quality indicators of SMOS were employed to filter data with lower quality: radiofrequency interference (Rfi), data quality index (dqx), and Chi2. Specifically, a similar approach to [27] (their Equation (4)) was used, where these three quality indicators were merged together to provide a unique quality index (Qi) ranging between 0 and 1 (with 0 being the highest quality). In this work, all the data with a Qi > 0.5 were considered as low quality, filtering all these data for SMOS, but also for DISPATCH and SMOS-BEC as they also used the same SMOS retrievals in their algorithms. Note that, in many cases, these low-quality data were already filtered in the own algorithms of DISPATCH and SMOS-BEC.

There are two main differences between DISPATCH and SMOS-BEC products: (1) the use of 1 km resampled SMOS brightness temperatures as input to the downscaling relationship in SMOS-BEC, and (2) the scale over which the downscaling relationship was calibrated (one SMOS pixel for DISPATCH and nine SMOS pixels for SMOS-BEC). For SMOS-BEC, [32] have shown that the larger the calibration spatial extent, the smoother the disaggregated SM image. Moreover, the sub-pixel (SMOS) SM variability in the disaggregated images was closely linked to the 1 km ancillary data used as input. In the case of SMOS-BEC algorithm, the strong relationship between SMOS SM and SMOS brightness temperatures at SMOS resolution involved a relatively larger weight (through the calibration coefficients) on the SMOS brightness temperature than on the MODIS data.

### 2.3. Land Use Data

The LU data used in this study were prepared from the vectorial information contained in the Andalusia Natural Heritage information system spatial database. This database was assembled at a detailed scale (1: 10,000) from geographic and alphanumeric information from the harmonized geometric integration of the natural habitat types of community interest, the LU information system, and vegetation maps in forest ecosystems. As this dataset contained polygons for more than 400 possible LU types, the first step was to gather these categories into a lower number (14). The coordinates were reprojected from UTM (EPSG:25830) to geographic coordinates (EPSG:4326) and finally, a regular

grid was prepared in the same grid as DISPATCH (1 km), conserving the information of the percentages of each vegetation cover in each pixel, as well as the dominant category.

For the LU and SM relationship shown in this work, only those pixels where the dominant LU covered more than 50% of the total area of the 1-km pixel were analysed. This avoids including pixels with too much heterogeneity and where the dominant LU only covers a small portion. Considering a threshold higher than 50% strengthened the robustness of the results but at the cost of reducing the number of pixels used. After different sensitivity experiments (not shown), the 50% threshold was retained since it included an acceptable compromise between a large enough surface of dominant LU and a large enough number of pixels fulfilling this condition in the area of interest, leading to the same conclusions as using a higher percentage.

### 2.4. API Model Data: Rainfall and Temperature

A simple antecedent precipitation index (API) model (Equation (1)) was used to simulate the SM according to the rainfall. This model was previously used in several works for different applications, e.g., [27,44]. The model was forced with rainfall and temperature data from the SPAIN02 v5 reanalysis dataset [37–39], available at a resolution of 0.1°. This version of the API model was based on previous versions used in [25,44]. It simulates the SM in a thin layer close to the surface only with the information of the previous (antecedent) value of SM and with the amount of rainfall (*P*) at each specific time (*t*). The exponential decrease of the SM slightly depends on the air temperature; however, there is no dependency on the soil type or other surface particularities:

$$sm_{(t)} = [sm_{(t-1)}\, e^{-\Delta t/\tau}] + [\Theta_{sat} - sm_{(t-1)}][1 - e^{-P_{(t)}/h}], \tag{1}$$

where $sm_{(t)}$ is the simulated SM (m$^3$ m$^{-3}$) for each time step ($\Delta t$, one day here). The first term at the right side represents the exponential decrease of the SM, based on the previous SM value ($sm_{(t-1)}$) and on $\tau$, which impacts the exponential decrease of $sm_{(t)}$ depending on the air temperature (*T*) following Equation (2), derived experimentally in [44].

$$\tau = (-7e^{-5}\, T^4) + (0.006\, T^3) - (0.03\, T^2) - (9.5\, T) + 287. \tag{2}$$

The second term at the right side of Equation (1) considers the increase in SM due to the rainfall and taking into account the precedent SM and the maximum SM possible before the soil saturation ($\Theta_{sat}$), being *h* the depth of the considered layer (3.5 cm in this case, similar to the depth of satellites SM measurements). The model was initialised with a spin-up of 2.5 years, starting the simulation in summer 2012, when the SM values were close to its minimal value, i.e., the residual value of SM (0.02 m$^3$ m$^{-3}$). In this work, only the results for 2015 were used, which was the year of analysis.

Subsequently, the differences between each SBSM product and the SM simulated by the API in each pixel were calculated, representing the biases of the products (Equations (3) and (4)).

$$BIAS_{DISP} = SM_{DISP} - SM_{API}; \tag{3}$$

$$BIAS_{SBEC} = SM_{SBEC} - SM_{API}. \tag{4}$$

Although they were named as biases, we assumed that they contained information about the real SM of each pixel. That is, the same API model was run for all the pixels, but the biases were different depending on the LU of each pixel, which indicates that the data from the satellites provided information regarding the different water retention capacities associated with each type of vegetation. We did not interpret the absolute value of the bias, as the two SM time series compared (SM from API model and SM from satellites) were not re-scaled for that aim. We only compared the values of the biases between different LU categories. As an example, we did not try to determine whether

DISPATCH overestimated the SM absolute values over the savanna pixels, but to determine whether the biases of the savanna pixels were larger or smaller than those of other LU types, e.g., grass.

### 2.5. Soil Type Data

The 16-category soil type database from the United States Department of Agriculture (USDA) used in the Weather Research and Forecasting (WRF) model [40] at a 3-km resolution was used to avoid the SM comparison between pixels situated in regions with different soil types. In this sense, the analysis was performed exclusively over those pixels characterised by loam, which is the dominant soil type in the whole region of analysis. This database was re-gridded to the same grid as DISPATCH (1 km), as done for the other data.

## 3. Results

The results were divided into two sections according to the two scientific questions. In the first section, the relationships between the satellite-derived SM and LU were analysed, and in the second section, the satellite SM at the pixel level was compared with in situ measurements at the experimental site.

### 3.1. Land Use and Soil Moisture Relationship

The dominant LU of Andalusia is shown in Figure 1a. Dense or open savanna covered 13% of the surface of Andalusia, while 25% of the surface was covered by fruit-tree crops (mainly olive trees) and 22% by grass. While the savanna dominated the northern and the elevated areas of the region, the grass dominated over the areas situated at lower altitudes. The grass is normally dry, yellow, and without photosynthetic activity in summer. In this work, we focused on the area situated in *Sierra Morena*, delimited with a yellow rectangle in Figure 1a and zoomed in Figure 1b. Since savannas are a composite of trees and grass in variable proportions, special attention was given to three well differentiated LU categories: dense savanna, open savanna, and grass, which represent approximately 27%, 12%, and 3% of this region, respectively. The *Santa Clotilde* site is marked with a yellow circle approximately in the centre of the area (Figure 1b). The dominant LU classification for this pixel was dense savanna (87% of dense savanna and 12% of open savanna, subgrid variability not shown).

Subsequently, the SM at different land covers was investigated from the analysis of the biases of the two SBSM products (DISPATCH and SMOS-BEC) with respect to the API SM. Figure 3 shows these biases found for the different LU categories for DISPATCH and SMOS-BEC. These results were divided for summer and winter periods, in order to detect SM differences due to the variable state of the vegetation and its influences on the upper layer of the soil.

The dense savanna, open savanna, and grass pixels exhibited different biases for DISPATCH (upper figures, Figure 3a,b), which were interpreted as different SM contents: the denser the proportion of trees, the higher the SM values. Conifer forest pixels showed higher SM contents, while other LU categories exhibited higher SM variability. These differences were conserved during the summer (right), although the values were smaller than for the winter, which was expected due to the smaller SM values in the summer in the region of analysis (see later in Figure 4). In the summer, evergreen forests showed higher SM than dense-savanna areas, while their SM was similar in winter. This particularity could be related to the fact that the dense savanna, although *dense* was also composed by grass between the trees, which could favour the evaporation from the soil, especially in the summer season when the grass is dry. This phenomenon did not occur in the forest as it was covered by more trees without open areas; phenomenon also observed for the conifer forests, preventing soil evaporation. In winter, however, the grass was green and active, which increased its capacity for water retention. There was an added effect leading to less SM beneath trees caused by the interception of light rain by the canopy in winter. Similarly, the difference in density between the two categories of shrub seemed to have a larger impact on the SM in the summer, with more water retention for denser shrubs. Non-irrigated fruit trees (mainly olive trees with a considerable surface of bare soil)

also exhibited different behaviour in the summer and winter, with a remarkable dryness in the summer in comparison with other LU.

These differences are more difficult to observe for SMOS-BEC (Figure 3c,d; note the different y-axis range with the upper figures). Nonetheless, the signal was similar to that for DISPATCH in the case of grass, open savanna, and dense savanna in the winter but different for the rest of the LU categories and for all the categories in summer. This could be related to the fact that the SMOS-BEC algorithm spatially smoothed the SM values due to the fact that the downscaling relationship was calibrated with several SMOS pixels. The inclusion of SMOS brightness temperatures (via their re-sampling at 1 km resolution) in SMOS-BEC systematically led to some spatial smoothing of the SMOS-BEC disaggregated images.

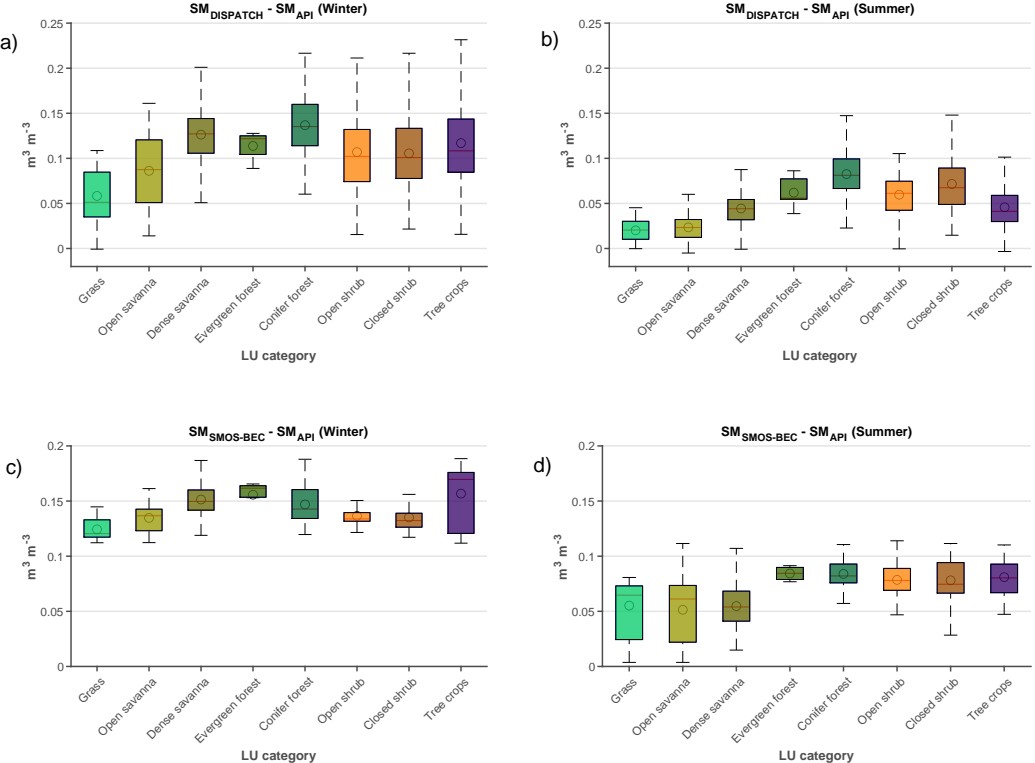

**Figure 3.** Soil moisture (SM) difference between Disaggregation based on Physical And Theoretical scale Change (DISPATCH) (**a**,**b**) or SMOS-Barcelona Expert Center (SMOS-BEC) (**c**,**d**) and antecedent precipitation index (API) (that is, the bias of the satellite-based soil moisture (SBSM) products is represented with respect to the modelled). The results are presented for different land use (LU) categories in the region of analysis shown in Figure 1. The boxplots provide information about the mean (circle marker), the median (red line), the 50% of the central distribution of the data (central black box) and the rest (whiskers) calculated using the mean SM biases during the winter (**a**,**c**) and summer (**b**,**d**). Colours correspond to the legend of LU categories in Figure 1. The absolute values provided should not been taken into account since the in situ and the satellite time series have not been rescaled for that aim. Only the differences between the LU categories should be considered.

### 3.1.1. Evolution of SM Differences through the Year and Comparison with the In Situ Data

We investigated in more detail how the differences in SM shown in Figure 3 evolved through the year under different conditions. This was shown in Figure 4, where the SM differences between dense savanna and open savanna (red lines) and between dense savanna and grass pixels (blue lines) are represented for DISPATCH (a) and SMOS-BEC (b) data. We checked whether the evolution of these differences was also observed with the in situ SM measurements at the *Santa Clotilde* experimental site

(see yellow circle in Figure 1b). Black dashed lines in Figure 4a,b show the SM differences between the measurements beneath the tree and at the grass site.

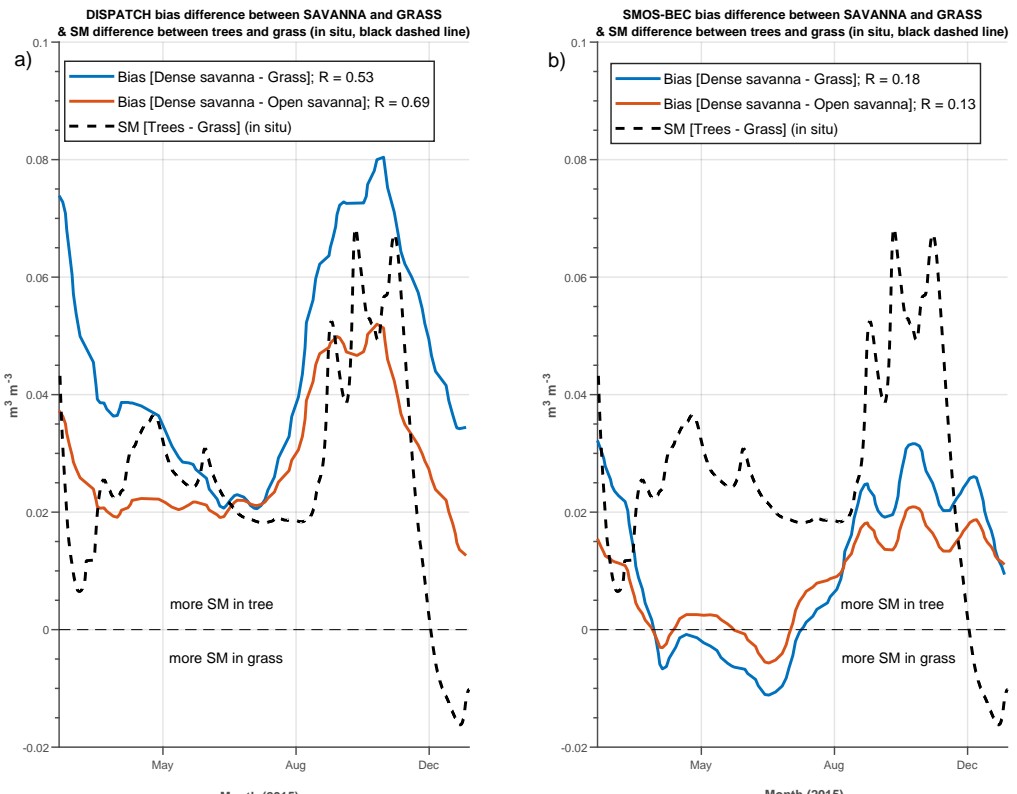

**Figure 4.** Mean soil moisture (SM) bias differences between dense savanna and grass pixels (blue lines) and between dense savanna and open savanna (red lines) for DISPATCH (**a**) and SMOS-BEC (**b**) through 2015. The black dashed line shows the evolution of the SM differences between the trees and the grass sites measured with the in situ probes, included as the reference for the comparison of the evolution (the differences of SM biases are represented with the red and blue lines while we show the evolution of SM differences with the black dashed line; therefore, the absolute values are not comparable, but the evolution is). The correlations between the red/blue lines and the black one are included in the legend, in order to indicate the ability of these products to represent the seasonal evolution of these differences.

In general, the upper layer of the soil beneath the tree retained more SM than the open area with grass, but not under all possible conditions. In summer, the differences remained almost constant, but always with higher SM content below the trees, where the tree shade had an important role to reduce the solar radiation, diminishing the evaporation from the soil. These SM differences between the tree and grass increased in the rainy season from September to mid-November, especially immediately after rain events (see rainfall in Figure 5). The more rapid evaporation of the rain from the grass surfaces led to higher SM contents beneath the trees under wet conditions. After the rainy months, in this particular year, only a few light rain events were observed from the second week of November to the end of December (see rainfall in Figure 5).

During this period, the SM decrease was dampened at the grass site but not beneath the trees (note the light rainfall and the different SM decrease from mid-November to the end of December in Figure 5a). In fact, the SM at the tree site was not affected by the light rain events. We think that this could be caused by the effect of the interception of the tree canopy, preventing the rain water from reaching the ground level, and thus its infiltration up to the 10-cm depth of the sensor. However, the light-rain effect was observable in the SM measured at the grass site due to the absence of trees (no interception, see the different decrease in Figure 5a in mid-November), finally leading to a higher

SM content at the grass than below the trees. While the light-rain effect was observed at 10 cm, it was not observed at 30 and 50 cm (Figure 5b,c).

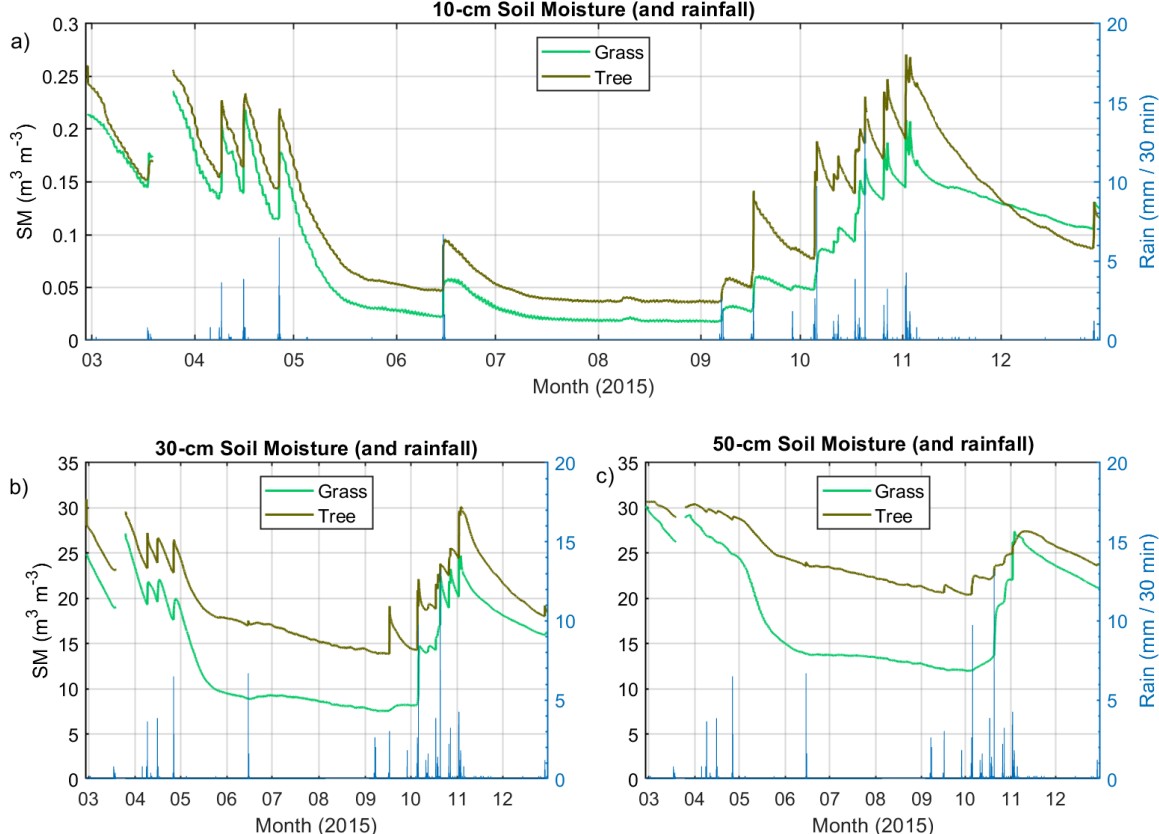

**Figure 5.** Soil moisture (SM) at the grass (light green) and tree (dark green) at the savanna site in *Santa Clotilde*: 10 cm (**a**), 30 cm (**b**), and 50 cm (**c**). The rainfall (mm/30 min) is represented with blue bars.

Other phenomena could contribute to these differences. The grass was green and active due to the relatively abundant rain of the previous weeks, favouring water retention due to the more active and numerous roots at shallower depths. Although there was also grass beneath the trees, the density was normally lower than at the open areas. Furthermore, the evaporation from the soil was not as important as during stronger rain events (high SM content in the surface) or during the summer (high solar radiation). It was also possible that the dew deposition (common in winter mornings with fair weather) could have an impact on the non-rainy period of December (with extra water supply in the open areas (lower minima temperatures) but not below the trees). An analogous explanation can be applied to the first days of March 2015, where the SM decreased faster beneath the trees than in the grass (see Figure 5a), under similar conditions to November–December 2015.

In contrast, the SM decrease was faster at the grass site than beneath the tree in summer, when the grass was yellow, dry, and inactive, with less capacity of water retention, exposed to stronger solar radiation (increasing evaporation), and with higher nocturnal temperatures avoiding the dew processes. In this case, the shadow of the trees in summer also seemed to have an important role in maintaining more SM than in the grass, while the interception effect was not present due to the absence of rain. These particularities, together with the longer dry season without rain led to the SM convergence to the residual water content of the soil both beneath the tree and in the grass (diminishing their differences).

The comparison of the blue and the red lines in Figure 4a shows how the differences in the biases for dense savanna, open savanna, and grass also increased during the rainy period around October–November, as also observed with the in situ data (Figure 4 black dashed line). The tendency

of the black dashed curve in November–December was also observed from the satellite products, although without reaching a moment with more SM in the savanna than in the open areas. The curves of the satellite products also indicate a higher capacity of the grass for water retention when it was active and green, as seen with the in situ data. The SM difference tendencies were also well-captured in spring and summer from the satellite data.

Although the similarity between the ground-based and satellite curves could be related to the fact that the biases (and also their differences) can increase with higher SM values during the rainy season, the relatively good values of the correlations indicate that the seasonal differences of these contrasting land uses can be detected with DISPATCH, with correlation values of 0.53 for the case of dense savanna minus grass and 0.69 for the dense savanna minus open savanna. As the correlation is higher for dense and open savanna differences, this could indicate that the local measurements beneath the tree and the grass better represent the dense savanna and open savanna LU categories, but this aspect was difficult to investigate as it depends on many other specific local particularities of the ground measurements.

On the contrary, although SMOS-BEC (Figure 4b) also showed some tendency to increase the differences between biases values in the rainy season and it followed some of the seasonal tendencies, in this case, the correlations between the curves were significantly lower (0.18 and 0.13). In some cases, the evolution of the differences in bias did not agree with the observations from the ground measurements. This could be related to the lowest capacity of this product to detect small-scale differences (and therefore LU differences) due to the horizontal smoothing applied, as commented in the analysis of Figure 3c,d and in Section 2.2.

### 3.2. Satellite Data at the Specific Pixel Versus 'In Situ' Data

From the previous section, we concluded that the satellite SM data provided useful information at the regional scale. In this section, we investigated whether the data at the pixel (local) scale could also be useful to investigate land–atmosphere interactions.

Although the evaluation of SBSM products was not the main objective of this study, it was necessary to deepen in the investigation of some of the scientific questions addressed. The more direct strategy to evaluate the SBSM products was to compare the pixel where the experimental site was located with the in situ data (Figure 6a,b and information about scores within the legend). The correlation was similar for DISPATCH (0.53), SMOS-BEC (0.57), and SMOS (0.59), evaluated with the measurements beneath the trees (Figure 6a) and in the grass (Figure 6b). The biases were not included since the two compared time series (SBSM products and in situ) have not been re-scaled for that aim.

The noise in the SBSM measurements is well observed in these graphics. Consecutive measurements show SM variations that are not real, which is a well-known limitation of the SM measurements from satellite. Indeed, some studies have shown how the information averaged from several consecutive SMOS satellite measurements often offers better results than accepting and using only one measurement at a certain time (see for example [27] or [45], where the results of their application were better using five or six SMOS passages). In this context, the evaluation of the two SBSM products was performed by applying a low-pass Butterworth filter with a cut-off frequency of several days (Figure 6c,d), in this case using 3 days (ideally equivalent to six SMOS measurements approximately). The correlation with the in situ data improved significantly up to values of 0.69, 0.67, and 0.72 for DISPATCH, SMOS-BEC, and SMOS, respectively, in the case of comparison with the measurements taken beneath the tree (Figure 6c), which implies improvements of 16%, 10%, and 13% from the original values.

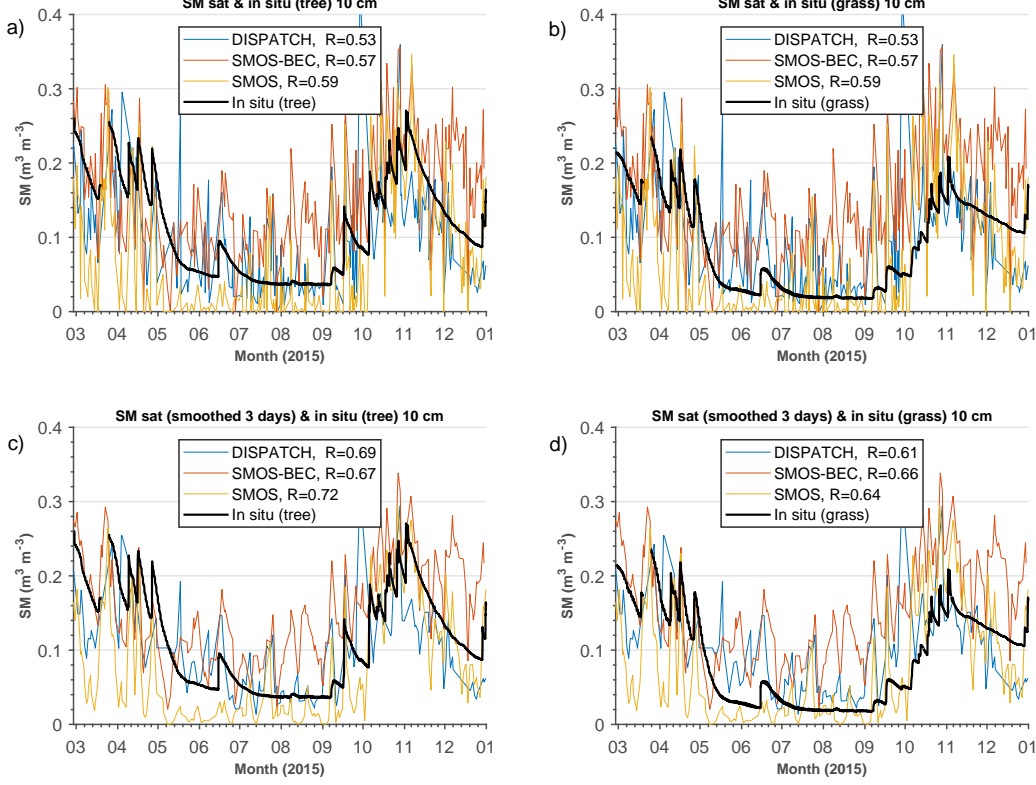

**Figure 6.** Comparison of soil moisture (SM) provided by DISPATCH (blue), SMOS-BEC (red), and SMOS (yellow) and 10-cm in situ measurements (black line), using measurements beneath the trees (**a**,**c**) and in the grass (**b**,**d**). Results with original data (**a**,**b**) and applying a low-pass filter of 3 days (**c**,**d**). Correlations between the SBSM products and the in situ measurements are indicated in the legend.

Although in some cases this strategy can reduce the SBSM products information from real rainy events, the reality is that it is difficult to disentangle these events from the noise of the signal. The effect of the rain events in the SM is normally appreciable in the measurements for several days, which encourages the use of various SMOS measurements. The optimum number of days to smooth the SBSM signal will depend on the specific objectives of the application and on the length and characteristics of the studied period, since smoothing the SM risks removing some information from rainy events. Figure 7a shows the correlation improvement when filtering the signal with a different number of days. This figure shows how the correlation improved significantly by simply applying a filter of a few days. For this specific evaluation of a long period (almost a year), the correlation continued to improve when filtering measurement oscillations of several days, since the general tendency of the SM is better captured.

Some of the noise of the SM signal could be related to differences between the ascending and descending retrievals. Therefore, we investigated if the correlation improved when considering a unique type of orbit of SMOS (Figure 7b,c for the descending and ascending orbits, respectively). The results indicated that the DISPATCH evaluation was significantly better when using only the ascending orbit of SMOS (blue line). The same was also observed for SMOS but with less improvement, while the differences between orbits were more difficult to observe for SMOS-BEC.

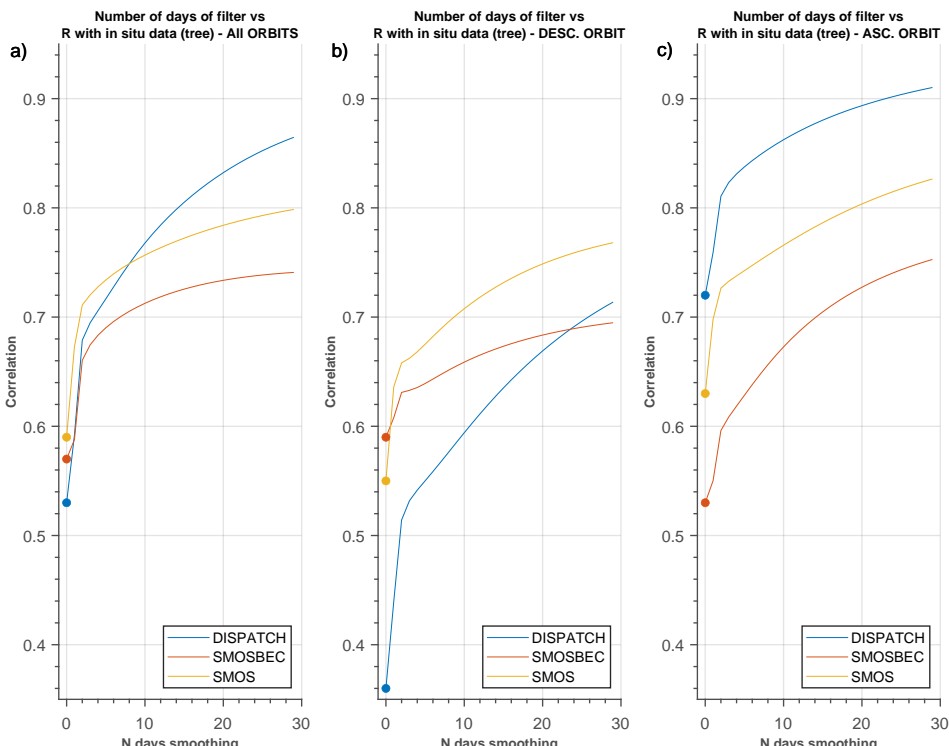

**Figure 7.** Correlation (y-axis) between the in situ (tree measurements) and satellite-based soil moisture (SBSM) records using a low-pass filter of different frequencies (x-axis). DISPATCH (blue), SMOS-BEC (red), and SMOS (yellow) using all orbits (**a**), descending (**b**), or ascending (**c**).

Using SM from Satellite to Investigate Land–Atmosphere Interactions

These high-resolution SBSM products are very appealing for some applications investigating the interactions between the surface and air. In this context, we analysed whether the relations between SM and evapotranspiration (ET or latent heat flux, Le) found with in situ data were similar when using SM from satellite. Figure 8a shows the scatter plot of Le normalised by the net radiation (Le/Rnet) as a function of the in situ SM at 10 cm, using mean daily values calculated within an interval of 4 hours around midday. These scatter plots were widely used to determine if the activity of the vegetation of the analysed site was dominated by the SM (dry and transitional regimes) or by the radiative energy (wet sites with enough SM), based on the first works of [46]. The larger the slope of the points, the larger the dependency of ET on SM.

By analysing all the points of the scatter plot in Figure 8a (grey and coloured points), we see a group with low SM, corresponding to the summer period with a dry regime associated with low values of Le/Rnet, while the other points show a linear relationship between both variables, corresponding to seasons with varying SM. This all-year behaviour can be also observed using the satellite products (Figure 8c,e,g), with similar correlation between both variables: 0.57 for the in situ data, and around 0.50, 0.33, and 0.44 for DISPATCH, SMOS-BEC, and SMOS products, respectively. However, the interpretation of this comparison for the whole year is difficult, as different periods with varied particularities are merged. Thus, in semi-arid regions like the savanna, the relationship between these two variables can change significantly from one season to the other, as plants are typically well adapted to long dry periods followed by rainy ones.

For this reason, the analysis was separated into three different and interesting periods: (1) a drying period in late spring (May); (2) the first rainy period after the summer (October); and (3) a drying period with few and light rain events after a considerably wetting of the area (November). The evolution of the SM and Le/Rnet is provided on the right side of Figure 8. The objective of this figure is to compare the results obtained from the SBSM products (Figure 8c–h) with the information provided by the in

situ data, considered as the reference (Figure 8a,b). In these plots, the SM data from the satellite have been smoothed using a 15-day filter, following the results obtained in Figure 7.

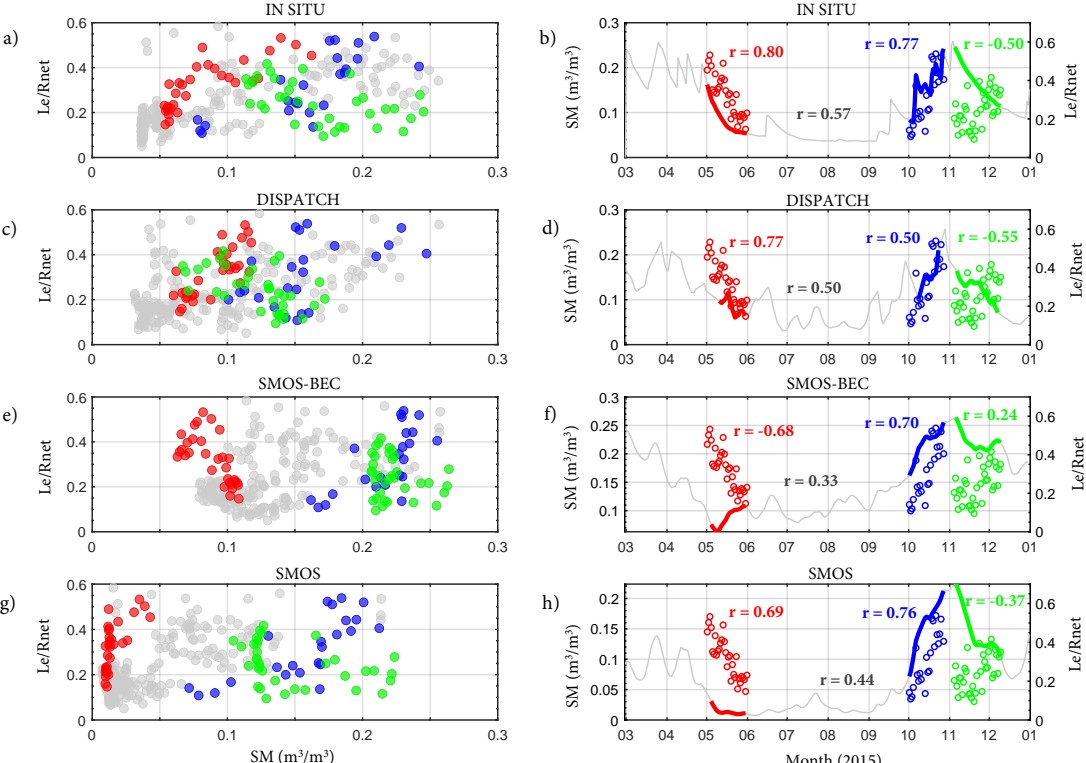

**Figure 8.** (**a**) Scatter plot of latent heat flux normalised by net radiation (Le/Rnet, y-axis) versus the in situ soil moisture (SM, x-axis) for daily values calculated from 10:00 to 14:00 UTC. Grey points correspond to all the analysed period; red points to the drying period in May; blue points to a wet period in October after the dry season; green points correspond to a drying period with only light rain events after the previous wet weeks. (**b**) In situ SM evolution throughout the year (lines) and Le/Rnet measurements (points), with the same periods than in (**a**) indicated with colours. Correlations between both variables are indicated with text. (**c**,**d**) Idem but using SM from DISPATCH at the pixel where the site is. (**e**,**f**) Idem but using SM from SMOS-BEC. (**g**,**h**) Idem but using SMOS measurements.

1. **The drying period in late spring (May, red):** This period is indicated in red and corresponds to the drying observed after the last rain event of the rainy season. The Le/Rnet decreases progressively as does the SM measured at 10 cm (Figure 8a,b). It clearly represents the so-called *transitional zone* where the plant activity is limited and dominated by the amount of SM. The SM decrease is well observed with the smoothed values of SM from DISPATCH and SMOS (Figure 8d and h), finding Le/Rnet and SM correlations that are similar to those observed with the in situ data (0.77 and 0.69 versus 0.80 with in situ data). However, SMOS-BEC was not able to reproduce the SM decrease in this period, showing an increase in SM during May (correlation −0.68). This was possibly caused by the smoothing applied in SMOS-BEC with the nearby SMOS pixels, which could include rain from other pixels to the analysed point.

2. **The start of the rainy season (October, blue):** This period is shown in blue in the graphics and corresponds to the progressive wetting associated with several consecutive rain events in October. In this case, there was also a clear response of the plants to the water input, increasing their activity as more water is available for photosynthesis, with a positive correlation of 0.77 between ET and SM (Figure 8a,b). The plant is here again in the *transitional zone* in which the evapotranspiration is

directly linked to the SM content. This relationship is also observed when using the SM from the three SBSM products (Figure 8c–h). The correlations found using SMOS and SMOS-BEC agree relatively well with those found with the in situ data.

3.  **The drying period without water limitation (November, green):** This is a period with few and light rain events, associated with the progressive drying of the soil but starting from a relatively humid soil in November (shown in green in Figure 8) and without the strong evaporation of other months due to the smaller values of radiation during this period of the year. The correlation between the Le/Rnet and SM was low and even slightly negative according to the in situ measurements, which indicated that the measured ET had no relation with the SM; the Le/Rnet continued to increase despite the SM decrease (measured at 10 cm). Comparing the in situ figures (Figure 8a,b) with the one obtained from satellite (Figure 8c–h), the three SBSM products were able to correctly represent this decrease in SM, obtaining a similar type of graphics and correlations. However, a weak SM increase was observed in mid-November from the SBSM products and not from the in situ data. Some light rain events were observed at the site, slightly affecting the 10-cm SM measured at the grass but not beneath the trees (due to the tree interception). As the SBSM products measure the SM at the upper layer of the soil, these signals in the SBSM products can be due to the differences in depth between the in situ and the satellite data. Therefore, in these cases when the rain is not intense enough to infiltrate deep in the soil, the SBSM products provide additional information about the most superficial layer that can be useful to relate with the transfers between the surface and the atmosphere. The increase in the Le/Rnet in mid-November was likely caused by the direct evaporation from the soil under these conditions.

At this point, it should be remembered that the good agreement of the satellite data in comparison with the in situ results was obtained after the application of a low-pass filter, which removed the SM oscillations of 15 days, producing smoothing in the SM signal from the satellite. The results obtained without applying any filter (i.e., using the SM signal directly from the SBSM products) did not produce the same agreement (Table 2), due to the reported high noise of the SBSM signal. This result encourages the post-processing of the SM signal for this type of study. The fact that these SBSM products can include information from rain events taking place at other locations within the 40-km area of SMOS (the main SM information in the three products) is another potential limitation of them.

**Table 2.** Correlation between soil moisture (SM) and latent heat flux normalised by net radiation (Le/Rnet) for the different analysed periods using the in situ data (considered the reference) and SM from the different satellite-based soil moisture (SBSM) products, where the results are provided after applying a low-pass filter of 15 days (as in Figure 8, first values of the table) and without any smoothing of the SM signal (second value in the table). Note the better agreement with the in situ values when applying the smoothing.

|  | **All Period** | **Drying** | **Wetting** | **Drying from Wet** |
|---|---|---|---|---|
| **In situ** | 0.57 | 0.81 | 0.77 | −0.50 |
| **DISPATCH** | 0.50/0.28 | 0.77/0.51 | 0.50/−0.33 | −0.55/−0.50 |
| **SMOS-BEC** | 0.33/0.34 | −0.68/−0.70 | 0.70/0.63 | −0.24/0.12 |
| **SMOS** | 0.44/0.40 | 0.69/0.49 | 0.76/0.70 | −0.37/0.09 |

## 4. Summary and Conclusions

The scientific and practical interest in SM has increased over the last few decades. This variable is key for many physical processes of the Earth and remains a challenge to measure and study. Several satellite missions were launched some years ago with the objective of providing SM information at the global scale with a resolution of around 40 km [22,23]. During recent years, new high-resolution satellite-based soil-moisture (SBSM) products have been developed, downscaling the coarser resolution up to 1 km or less, e.g., [29,31]. These SM products at high resolution are very attractive for

many other applications, including the analysis of land–atmosphere interactions [47], agriculture management [28,48], and ecological studies [49] of different vegetation types.

In this work, we investigated if these SBSM products can be used for two specific applications. On the one hand, we studied whether the differences in SM due to different vegetation cover could be detected from these high-resolution products (DISPATCH and SMOS-BEC products were analysed here, both at 1-km resolutions). To this aim, the SBSM products were compared to the expected SM simulated with a simple API model forced with the real rainfall. This step was needed to remove the SM differences due to the heterogeneity in precipitation. These differences (biases) were analysed for each LU type and related to the ability of water retention of each vegetation type. We found SM differences related to the tree cover, with more SM in those pixels covered by a higher percentage of trees (in this order: forest, dense savanna, open savanna, and grass), as was expected and observed in previous studies, e.g., [50]. These differences were analysed in detail through 2015, in order to investigate their evolution under different seasonal conditions. To check these findings, we compared these differences in SM with those found for in situ measurements beneath a trees and in grass.

The results found from the regional analysis using satellite data were similar to those obtained from the local analysis using in situ data, even for the evolution of these differences under particular conditions of the year. For example, some light rain events in November–December 2015 led to a different SM evolution in the grass in comparison to the tree site. This was likely caused by the interception effect of the canopy [51], which prevented part of the rain from reaching the ground beneath the trees, while the solar radiation effect was not as strong as in summer [52]. This phenomenon had an influence on the SM differences between the sites, which was also captured from the satellite data.

The analysis of these processes demonstrated how the water retention in the upper layers of the soil was, in some cases, highly influenced by the type and seasonality of the rainfall, which may have important implications for the SM differences found between different land cover types. The effect of the tree canopy is also different depending on the season. In summer, the shadow of the tree reduces the solar radiation from the sun, diminishing the evaporation and leading to more SM beneath the tree. In winter, it is the canopy interception which leads to lower SM content below the tree. But the latter effect also depends on the type and intensity of precipitation and antecedent humidity conditions, in accordance with results found by [53] in similar ecosystems. Although, in this work, we focused on the upper layer of the soil, understanding all these processes is crucial to investigate how the ecosystem can react to the different and variable particularities of each season, which also highlights the delicate influences that regional LU changes can have over the eco-hydrological cycle and the ecosystem functioning of some areas [54]. In this paper, we demonstrated that this type of analysis can also be carried out using satellite data. These results were more robust when using DISPATCH in comparison to SMOS-BEC, as for the latter, a spatial smoothing was applied, which is appropriate for other applications but not to study LU–SM relationships.

In the second part of this work, three SBSM products (DISPATCH, SMOS-BEC, and SMOS) were evaluated at the pixel-scale by comparing them with the in situ SM data, in order to determine if they could be used instead of the ground-based measurements. This evaluation indicated how applying low-pass filters of several days considerably improved the correlation with the in situ data. The cut-off frequency of the filtering depended on the purposes of the study; however, significant improvements are observed when filtering oscillations of around 3 days. Finally, we evaluated if the relationships of SM and Le/Rnet could also be studied using satellite data. To do this, the results using in situ data were compared to those obtained using SM from the products.

The results suggested that it was possible to use the SBSM products at the specific pixel of the site after filtering the high-frequency oscillations. This could be a great advantage in sites without in situ SM data. In certain cases, the SBSM products have the advantage of providing information from the upper layer of the soil that is not appreciable from deeper measurements under conditions of light rain not able to penetrate enough in the soil. However, these products can also include in their SM signal

information from rain events that did not occur at the specific point but in the area of the MW satellite (40 km), as well as from farther pixels if a spatial smoothing is applied, as in the case of SMOS-BEC. This downscaling issue can be more important in regions with heterogeneous rain events, for example, during local convective storms. Regarding the ET measurements, in this work we did not separate the ET from the tree and grass sites, which could be an interesting aspect to investigate in the future.

We attempted to determine whether we could use high-resolution SBSM products to investigate SM differences due to contrasting vegetation and to study land–atmosphere interactions. Based on the results, the answer was yes, at least in semi-arid regions similar to the one analysed here. However, some signal processing is recommended prior to analysis (smoothing of data), as well as an assessment of the quality of the data. The limitations of this methodology were those related to the horizontal resolution of the different data used. Therefore, these results encourage making efforts to improve the resolution of SBSM products to some hundreds of meters, as well as reducing the noise of their signal. By doing this, many other interesting applications will appear, including the remote monitoring of the water content in different landscapes, crops, and even irrigated fields as it will be easier to find pixels completely covered by specific land uses at those resolutions. Thanks to the long time-series of SM already available, the study of the interannual variability and climate-change or land-use change effects on the SM at high resolution is also possible now.

**Author Contributions:** C.R.-C. developed the main strategy, carried out the calculations and wrote the article. M.L. and F.L. contributed to the development and brain storming of the main idea. N.O. and O.M. prepared the DISPATCH data for the studied region. M.P.G.-D. and A.A. managed and prepared the in situ data at the *Santa Clotilde* site. D.A. developed the LU maps for the region of analysis from SYPNA data. Thierry Pellarin developed the API model. R.C.S., R.D.-D., O.H., A.B. and C.Y. facilitated other in situ data finally not used in this work for space reasons and contributed to the scientific discussion of the results. All the authors contributed to the improvement of the manuscript text. All authors have read and agreed to the published version of the manuscript.

**Funding:** Carlos Román-Cascón work was funded through a Postdoctoral Grant funded by the Centre National d'Études Spatiales (CNES). The European Commission Horizon 2020 Programme for Research and Innovation (H2020) in the context of the Marie Sklodowska-Curie Research and Innovation Staff Exchange (RISE) action (ACCWA project, grant agreement No 823965) has funded the PhD of Nitu Ojha. Dr. Andreu's work was funded by the European Union's Horizon 2020 Research and Innovation programme under the Marie Skłodowska-Curie grant agreement No 703978. The Spanish government project CGL2015-65627-C3-3-R (MINECO/FEDER) has also partially funded part of this work.

**Acknowledgments:** The authors thank AEMET and UC for the Spain02 v5 dataset, available at http://www.meteo.unican.es/datasets/spain02. We also acknowledge the ICTS-EBD technical resources. SMOS-BEC data were produced by the Barcelona Expert Center (bec.icm.csic.es), a joint initiative of the Spanish Research Council (CSIC) and the Technical University of Catalonia (UPC), mainly funded by the Spanish National Program on Space. The authors would like to thank the SMOS CATDS L3SM product (http://www.cesbio.ups-tlse.fr/SMOS_blog/), IFAPA for the Sta Clotilde data, SYPNA land use information data, and the soil-type data available in the WRF model from USDA.

**Conflicts of Interest:** The authors declare no conflict of interest.

## Abbreviations

The following abbreviations are used in this manuscript:

| | |
|---|---|
| BEC | Barcelona Expert Center |
| DISPATCH | Disaggregation based on Physical And Theoretical scale Change |
| ECMWF | European Centre for Medium-Range Weather Forecasts |
| LST | Land surface temperature |
| LU | Land use |
| MODIS | Moderate Resolution Imaging Spectroradiometer |
| MW | Microwave |
| NDVI | Normalised differences vegetation index |
| SBSM | Satellite-based soil moisture |
| SEE | Soil evaporative efficiency |
| SM | Soil Moisture |

SMOS      Soil Moisture Ocean Salinity
SYPNA     Information Systems about the Natural Resources of Andalusia

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
