# Peer review of "Can We Use Satellite-Based Soil-Moisture Products at High Resolution to Investigate Land-Use Differences and Land–Atmosphere Interactions? A Case Study in the Savanna"

_remotesensing, doi:10.3390/rs12111701_

Round 1

Reviewer 1 Report

Review of Can we use Satellite-Based Soil-Moisture Products at High Resolution to Investigate Land-Use Differences and Land-Atmosphere Interactions? A case study in savanna by Román-Cascón et al. in Remote Sensing

Summary of study

In this study, the authors used two high-resolution satellite-based soil moisture (SBSM) products to compare variation in SM across different land cover types, and to test their suitability for investigating land-atmosphere interactions. The products, which were created previously through statistical downscaling of coarse-resolution remote sensing SM, are here applied to investigate a semi-arid region in southern Spain. To ensure they were examining spatial variation in SM due to land cover and not due to precipitation (P), the authors used a model to estimate spatial variation in SM due to P, and then subtracted this from the SBSM products. The authors found SM in the summer months tended to be higher under ecosystems with higher tree density, such as conifer and evergreen forests, relative to shrubs, savannas and grasslands, highlighting an important role of trees in shading the land surface and so helping to preserve ecosystem moisture. Different behaviour occurred in winter, however, with dense savanna showing SM that was comparable with that under forests. The authors suggest this was due to high soil evaporation over photosynthetically inactive grasses in the summer leading to faster moisture depletion, while in winter the presence of grasses contributed to SM retention. Similar results were found when the authors compared in situ SM measurements taken under a tree, and in open grassland over the course of a year. SM from both SBSM products correlated well with SM measured under the tree and in the open grass, especially when smoothing was applied. However, the SMOS-BEC product showed much smaller differences between different land cover types. Finally, the authors tested whether SBSM could replace in situ SM data to analyse land-atmosphere interactions, and showed some degree of success, though performance varied between products.

The paper contains some very nice science and analyses and I think it will be of interest to a wide audience who might want to use SBSM products in various applications. For these reasons, I think that the study is worthy of publication. However, the manuscript as it stands can still be improved upon. In particular, it was quite challenging to read in places, and I think the structure could also be made more logical. I have made a few general suggestions, followed by detailed line-by-line corrections for English and grammar.  

General comments

The abstract didn’t do a very good job at summarising the main (and most interesting) conclusions of the study. The authors state the comparisons and hypotheses that were made, but the actual findings are hard to extract from the text. Could add a sentence along the lines of, “We showed that…” or “We found that…” to help highlight the main results. It would be good to mention that differences in SM were found to be related to tree cover, that SBSM products have potential to be substituted for in situ SM to study land-atmosphere interactions, and possibly the better performance of the DISPATCH product relative to SMOS-BEC.

The structure of the paper needs to be given a bit more consideration, and if the current structure is retained, the authors need to present a stronger argument for doing so by connecting the sections with some explanatory linking text that explains the logic of the progression. For example, would it make more sense for the comparisons against in situ data to come first, so the reader has an understanding of the quality of the SBSM products? Also the results section switches from analyses using the in situ measurements from under the tree and grass, to results based on land cover data, back to the tree and grass measurements. For me, it would make more sense to look at the smallest scale first, and then ‘zoom out’ to look at SM variation over larger scales. Furthermore, some of the methods were described in the results section, such as the difference between SBSM and the rainfall-based SM estimates. Thinking about and revising the paper structure could lead to a substantially improved manuscript that would be easier to follow and would properly showcase the main findings of the analysis.

The results of the study are presented, immediately followed by a summary and conclusions section, which does not fully contextualise the results in the context of the literature, or provide an in-depth discussion of the caveats of the study. For example, it would be good to know whether other studies have also found higher SM under forests, and in what context, and some discussion on what is known about evapotranspiration rates over forests, savannas and grasslands. Are the findings in line with what the authors had expected? Might have naively expected to find the opposite, i.e. lower SM under forests due to high water consumption by trees. How important is understory vegetation for ecosystem water retention? What are the implications of the difference between winter and summer? How is regional land-use change likely to be affecting the hydrological cycle? Furthermore, potential issues of downscaling could be commented upon.

Line-by-line remarks

Line 14 – ‘This evaluation highlights the potential of the SBSM products for this objective through this methodology” – at this point the reader has lost which objective and methodology you are referring to.

Line 24 – “water, energy, and carbon cycles

Line 39 – “This allows us to analyse…”

Line 40 – “the land-vegetation-atmosphere transfers” – transfers doesn’t sound quite right here, could rephrase to “how moisture transfers from the land to the atmosphere via vegetation”

Line 42 – determine

Line 43 – I disagree that 1 to 2 km is ‘typical’. Global circulation models which also need validating against in situ data can have resolutions on the scale of 100-200 km

Line 44 – “In this context, analyzing SM from space seems a good alternative to complement measurements performed on the ground”

Line 46 – satellites specifically designed for this aim

Line 50 – what do you mean by “commented studies” and “the mesh”

Line 52 – what resolutions? Be specific e.g. for resolutions on the scale of 1 to 2 km

Line 54 – ‘these aims’ not defined in this paragraph so state them again here for clarity

Lines 62-70 – combine with previous paragraph

Line 71 – remove the word “main” and maybe write your questions as questions? What do you mean by “appreciate” differences? Perhaps “identify” would be better here?

Line 72 – evolution through the year

Line 75 – the second question asks or addresses, not ‘analyses’

Line 80 – unclear how the proportion of land cover is important here

Line 81 – what do you mean by ‘each component’

Lines 78-94 – some of this description of the study site would be better in the Methods section

Line 101 – maybe analysed instead of confronted?

Methods and Data – write in the simple past tense not present tense and put study site section at the start

Line 103 – no reference given for the SPAIN-02 renalaysis

Lines 99-115 – this overview paragraph could be shortened and simplified – make it clear you are going into more depth below.

Table 1 – the caption is too brief. Can you explain the “Info” column more? Are these input datasets used to estimate SM? Give definitions of abbreviations, e.g. Le, Rnet etc. The information in Table 1 also doesn’t match what’s reported in the text – e.g. DISPATCH from MODIS LST or ECMWF LST? Make sure you double check all details are correct.

Line 131 – SMOS resolution (i.e. 1 km)

Line 136 –  Slight edits: “Note that since DISPATCH and SMOS-BEC have different grid coordinates, the 1-km grid of SMOS-BEC was interpolated to the same 1-km grid as DISPATCH, in order to compare both with the LU data prepared in the same reference grid.” Also mention what interpolation method was used to change the grid.

Line 140 – it’s a bit unclear what the coarse SMOS data was used for

Line 141 – re-gridded from what?

Line 145 – A similar approach to

Line 149 – ‘passages’ not right word here

Line 177 – where are the results of this sensitivity analysis shown?

Line 183 – “quite acceptable results” – this sounds a bit vague

Line 209 – proportions

Line 212 – “These measurements are taken over the Santa Clotilde experimental site (Fig. 2), the location of which is shown with a black circle in Figure 1b.

Line 219 – ‘This aim’ unclear what aim being referred to

Lines 219-229 mention EC flux tower – how tall? How many years of data?

Line 231 – Remove ‘As commented before’, here and elsewhere in the manuscript. Remove ‘main’

Line 232 – questions don’t ‘analyse’ or ‘investigate’. Please rephrase. E.g. In the first section, relationships between satellite-derived SM and LU are analysed, and in the second section, satellite SM at the pixel level are compared with in situ measurements.

Line 237 – ‘fruit-tree crops’

Line 239 – the Guadalquivir Valley is not marked – either add to map or remove reference here

240 – clarify what you mean by the grass being ‘inactive’ – i.e. no photosynthetic activity?

Fig. 1 caption – state that the yellow rectangle indicates Sierra Morena and black circle indicates Santa Clotilde

Line 243 – Well differentiated LU categories

Line 244 – the black circle is very hard to see! Can you make the edge thicker?

Line 246 – what do you mean ‘now shown’? Where do you get the sub-grid information from?

Line 247 – here and elsewhere you need to correct the tense of the text. Write in the past tense to explain what you did.

Ines 247 – 265 – Here you are describing methods, not results. OK to write a brief statement on what you did and then go on to explain what you found, but this level of detail would be more appropriate for the Methods.

Caption for figure 3 – explain abbreviations for vegetation types

Line 282 – similar to that for DISPATCH in the case of grass, open savanna and dense savanna

Line 285 – algorithm spatially smooths the SM values

Line 287 – Evolution through the year (not along the year – here and elsewhere in the text)

Lines 311-322 – the mechanisms for why SM declines more under trees than grass in winter could be better explained, as the text was slightly hard to follow here. How do you know that interception is causing the difference in SM decline and not higher water consumption by the tree? Does interception not normally occur at the leaf level?

Line 313 – Impact on

Line 300-332 Separate this paragraph into two

Line 333 – test hypothesis/ strengthen argument not strengthen hypothesis

Line 335 – can also be detected in the SBSM products (not appreciated)

Line 346 – similarity

Line 385 – what do you mean by SMOS passages?

Fig. 7 – could move to supplementary material

Line 401 – interactions between surface and air (not transfers)

Line 468 – reported high noise

Line 469 – this type of study

Line 473 – the last few decades

Line 477 – from hundreds of metre, to a few kilometres or even larger if considering using SM observations to evaluate climate models

Line 480 – Replace “in this sense” with “To address this issue, several satellite missions…”

Lines 473-488 include some references from the literature in this paragraph – e.g. studies on satellite SM products, evidence  that SM products are attractive for other applications, and for the limitation of the products

Line 489 – delete ‘In this sense”, make past tense

Lines 493 and 495 – repetition of “To this aim”

Line 495 – state again here why API model was needed – i.e. to account for the fact that variation of P in space is a strong control on spatial variation in SM and needs to be removed before other controls can be detected.

Line 499 – Through the year (not along)

Line 519 – replace ‘before doing it’ with ‘prior to analysis (smoothing of data), as well as an assessment of the quality of the data.’

Lines 521-522   - rephrase sentence

Author Response

Please, see the attached pdf file.

Reviewer 2 Report

Dear Authors: please see my attached comments.  Thank you.

Author Response

Please, see the attached pdf file.
